# Scoping review on interventions to improve adherence to reporting guidelines in health research

David Blanco,[1] Doug Altman,[2] David Moher,[3] Isabelle Boutron,[4] Jamie J Kirkham,[5] Erik Cobo[1]

¹Statistics and Operations Research, Universitat Politècnica de Catalunya, Barcelona, Spain
²Nuffield Department ofOrthopaedics, Rheumatologyand Musculoskeletal Sciences,Centre for Statistics in Medicine, University of Oxford, Oxford, UK
³Centre for Journalology, Ottawa Hospital Research Institute, Ottawa, Canada
⁴Centre d\'épidémiologie Clinique, Université Paris Descartes, Paris, France
⁵Biostatistics, University of Liverpool, Liverpool, Merseyside, UK

**Correspondence to**
David Blanco;
david.blanco.tena@upc.edu, david.blanco19@gmail.com

## ABSTRACT

**Objectives** The goal of this study is to identify, analyse and classify interventions to improve adherence to reporting guidelines in order to obtain a wide picture of how the problem of enhancing the completeness of reporting of biomedical literature has been tackled so far.

**Design** Scoping review.

**Search strategy** We searched the MEDLINE, EMBASE and Cochrane Library databases and conducted a grey literature search for (1) studies evaluating interventions to improve adherence to reporting guidelines in health research and (2) other types of references describing interventions that have been performed or suggested but never evaluated. The characteristics and effect of the evaluated interventions were analysed. Moreover, we explored the rationale of the interventions identified and determined the existing gaps in research on the evaluation of interventions to improve adherence to reporting guidelines.

**Results** 109 references containing 31 interventions (11 evaluated) were included. These were grouped into five categories: (1) training on the use of reporting guidelines, (2) improving understanding, (3) encouraging adherence, (4) checking adherence and providing feedback, and (5) involvement of experts. Additionally, we identified lack of evaluated interventions (1) on training on the use of reporting guidelines and improving their understanding, (2) at early stages of research and (3) after the final acceptance of the manuscript.

**Conclusions** This scoping review identified a wide range of strategies to improve adherence to reporting guidelines that can be taken by different stakeholders. Additional research is needed to assess the effectiveness of many of these interventions.

## BACKGROUND

Approximately 85% of all biomedical research today is estimated to be wasted, due, in part, to incomplete or inaccurate reporting.[1] The past two decades have given rise to a number of changes in an effort to help authors and the broader scientific community properly report research methods and findings, which would allow them to contribute to the broader goal of combating waste in biomedical research. The most prominent of these changes has

**Strengths and limitations of this study**

- ► We considered a wide range of reporting guidelines as well as their extensions.
- ► Merging the evidence found in the published and grey literature allowed us to provide a broad picture of how the problem of enhancing adherence to reporting guidelines has been tackled so far and could be faced in the future.
- ► The screening and data extraction were performed in duplicate.
- ► We could have missed evidence of possible interventions that may not be present in the published or grey literature but are instead used in practice and continue to be used.

been the inception of reporting guidelines (RGs) for different study types, data and clinical areas.[2]

The vast majority of RGs have not yet been assessed as to whether they help improve the reporting of research,[3] but some, such as the Consolidated Standards of Reporting Trials (CONSORT) for the reporting of randomised controlled trials (RCTs),[4] have been shown to enhance the completeness of reporting.[5 6]

Dozens of systematic reviews have explored the extent of adherence to some RGs in certain areas of health research.[7–10] Samaan *et al*[11] went one step further and performed a systematic review of systematic reviews assessing adherence to RGs. As they considered a broad range of clinical areas and study designs, their results provided a global picture of adherence to RGs in health research. Although some studies reported acceptable overall levels of completeness of reporting and found that it had improved since the introduction of certain RGs such as CONSORT, the authors of most of the reviews (43 of 50, 86%) concluded that more improvement is needed or that adherence to RGs was inadequate, poor, medium or suboptimal. Therefore, it is warranted to explore

Box 1    Relevant definitions in the context of this scoping review

**Adherence**

Action(s) taken by authors to ensure that a research report is compliant with the items recommended by the appropriate/relevant reporting guideline. These can take place before or after the first version of the manuscript is published.

**Endorsement**

Action(s) taken by journals to indicate their support for the use of one or more reporting guideline(s) by authors submitting research reports for consideration.

**Implementation**

Action(s) taken by journals to ensure that authors adhere to an endorsed reporting guideline and that therefore published papers are completely reported.

**Complete reporting**

Pertains to the state of reporting of a study report and whether it is compliant with all the items recommended by the appropriate/relevant reporting guideline.

and develop strategies to improve the current levels of adherence to RGs.

In recent years, several initiatives aiming to improve adherence to RGs have been proposed, some of which have already been evaluated. For example, the effect of journal endorsement of RGs[3 5 6] and the implementation of writing aid tools for authors such as the CONSORT-based web tool (COBWEB)[12] have been assessed. While some of these strategies have not been shown to have a benefit,[3] others report better but still suboptimal levels of reporting[5 6] or even clear benefits.[12 13]

As mentioned, several reviews have analysed the quality of reporting in different clinical areas and for different study types.[7–10] However, no scoping review has been performed that provides a global picture of different strategies aiming to improve adherence to RGs. Given the low levels of completeness of reporting in health research that have been observed,[11] along with the imperative need to take further actions for mitigating this problem, we considered that performing such a scoping review was warranted.

In addition to analysing the implementation and effect of interventions that have already been evaluated, we aimed to gather other possible strategies that could be implemented and evaluated in the future.

For clarification, some relevant terms used throughout the scoping are defined in box 1, which is based on Stevens *et al*.[3]

## METHODS

As presented in the published protocol,[14] this scoping review follows the methodology manual published by the Joanna Briggs Institute for scoping reviews.[15]

## Objectives

The scoping review questions are:
1. What interventions to improve adherence to RGs in health research have been evaluated?
2. What further interventions to improve adherence to RGs have been performed or suggested but never evaluated?

We aimed to analyse and classify the interventions found for both questions in order to obtain a wide picture of how the problem of adhering better to RGs has been tackled so far and can be tackled in the future.

## Eligibility criteria

We included:
1. Studies evaluating interventions aiming to improve adherence to RGs in health research, irrespective of study design.
2. Commentaries, editorials, letters, studies and online sources describing possible interventions to improve adherence to RGs that have been performed or suggested but never evaluated.

The RGs considered were those shown on 8 May 2017 on the Enhancing the QUAlity and Transparency Of Health Research (EQUATOR) Network website[16] as 'RGs for main study types'. In addition, we included Quality of Reporting of Meta-analyses, since it was the precursor of Preferred Reporting Items for Systematic Reviews and Meta-Analyses (PRISMA). Online supplementary file 1 shows all RGs considered.

We considered the following languages: English, Spanish, French, German and Catalan.

## Exclusion criteria

We have excluded references that include interventions that do not specifically aim to improve the completeness of reporting, even though these interventions may actually influence completeness. For example, we have excluded clinical trial registration even though it may enhance the completeness of reporting, because its main goals are to improve clinical trial transparency while also reducing publication and selective reporting biases.

## Search strategy and study selection

On 8 May 2017, we searched PubMed, EMBASE and Cochrane Library databases for articles published between 1 January 1996 and 31 March 2017, in accordance with our scheduled search.[14] The detailed search terms for PubMed can be found in the protocol.

The retrieved studies were exported into Mendeley and duplicates were automatically removed using it. One reviewer (DB) first screened the titles and abstracts for eligibility. Each of the other two reviewers (JJK and EC) was randomly assigned 50% of the references and screened the titles and abstracts independently of the first reviewer. The reviewers classified the references into one of the following groups:

A. Evaluated: Includes references describing interventions to improve adherence to RGs that have been empirically assessed.

B. Non-evaluated: Includes references describing interventions to improve adherence to RGs that have been performed or suggested but never evaluated.

C. Unclear: Includes references (1) containing vague statements such as 'authors, editors and journals have to adhere better to RGs to improve the quality of reporting' or 'greater efforts have to be made by authors to check that their research is compliant with (the relevant RG)', or (2) not having the abstract available.

D. Excluded: Includes references (1) not describing interventions to improve adherence to any of the RGs considered and (2) describing but not evaluating certain interventions that have already been classified as evaluated.

Disagreements were solved by discussion among the reviewers.

Second, one reviewer (DB) examined the full text of all group A and B references to confirm the previous classification, then all group C references to reclassify them either as group A, B or D. Reclassification was verified by the initial reviewer (JJK or EC). Finally, one reviewer (DB) ensured literature saturation by searching the reference lists of included studies, the lists of articles citing them according to PubMed, and the individual studies included in two relevant systematic reviews.[3 6]

In addition, we performed a grey literature search, which included: the websites of networks and organisations promoting the use of RGs (ie, EQUATOR Network and National Library of Medicine Research Reporting Guidelines and Initiatives); work groups of medical journal editors (ie, International Committee of Medical Journal Editors and World Association of Medical Editors); biomedical journal publishers (ie, BMJ Publishing Group and BioMed Central); funding agencies (ie, National Institute of Health and European Research Council); online platforms of postpublication peer review (ie, PubPeer and ScienceOpen); and the abstract books of the past editions of the International Congress on Peer Review and Biomedical Publication.

Some of the included references were described in studies coauthored by some of the authors this scoping review. These references underwent the same process of screening, data extraction and data synthesis as the others.

## Data extraction

A data extraction form was developed to collect the information necessary for data synthesis. Two reviewers (DB, JJK) independently performed a pilot data extraction on a random sample of five articles and subsequently refined the form.

Extracted data included:

1. Publication characteristics: title, year of publication, author, author's affiliation country and field of study.
2. Characteristics of the intervention:

a. Classification as evaluated or non-evaluated.

b. Research stage: education, grant writing, protocol writing, manuscript writing, submission, journal peer review, copyediting and postpublication.

c. Rationale of the intervention, which refers to the deduced reasons why the intervention is evaluated or proposed.

d. For evaluated interventions: details of the intervention, study design (eg, RCT, before–after, etc), RGs considered and format (checklist, bullet points and/or examples), period of intervention, number of journals and articles involved, effect size of the intervention on adherence to RGs and measure used to assess this effect.

3. Relevant conclusions.

Two reviewers (DB, JJK) independently performed data extraction for all studies except for the individual studies of the two systematic reviews evaluating journal endorsement of RGs,[3 6] since none of these studies described further interventions and their results had already been reported in these reviews. Discrepancies between reviewers were discussed and solved by consensus.

## Data synthesis

Following data extraction, interventions to improve adherence to RGs were categorised as follows:

1. Training on the practical use of RGs: mentoring of different stakeholders on the practical use of RGs.
2. Enhancing accessibility and understanding: dissemination of RGs and the improvement of authors' understanding of their content.
3. Encouraging adherence: suggestions and tools to facilitate compliance.
4. Checking adherence and providing feedback: checking the level of compliance and indicating incorrect or missing items.
5. Involvement of experts: interaction and cooperation on methodology and reporting.

One reviewer (DB) performed the initial categorisation, which was verified and refined by the other two reviewers (JJK and EC).

Furthermore, we determined the existing gaps in research on the evaluation of interventions to improve adherence to RGs. More specifically, we identified which categories of interventions and which research stages have not been addressed so far in studies evaluating interventions.

We did not perform a meta-analysis of the observational studies assessing journal endorsement of RGs that were not included in the two systematic reviews previously mentioned.[3 6] We considered that, for the purpose of this scoping review, these systematic reviews provided a reliable picture of the impact of this editorial intervention.

## Deviations from the protocol

In order to better capture the most relevant aspects of the included studies, the original data extraction form proposed in the protocol was modified. We removed

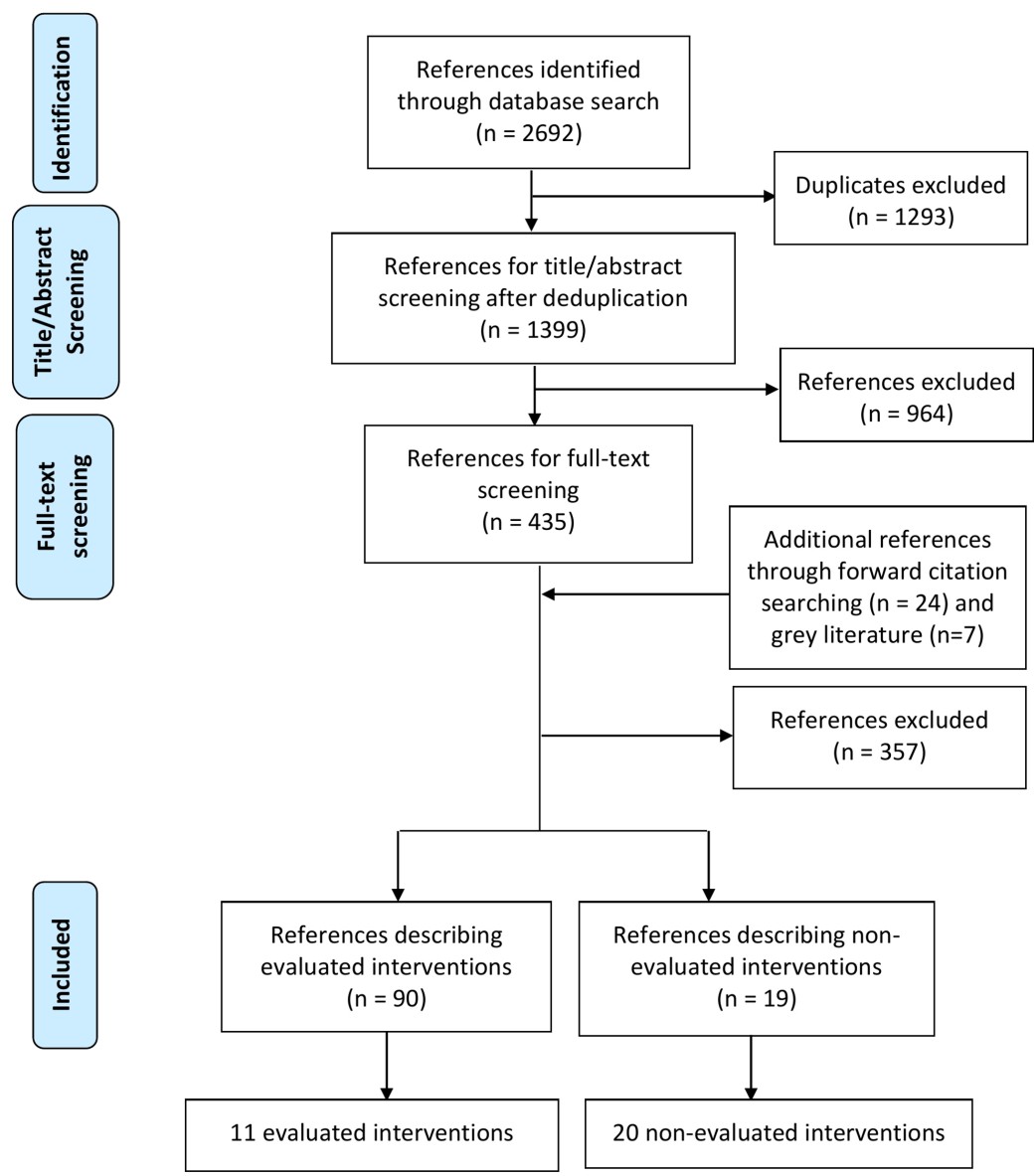

**Figure 1** PRISMA flow diagram. PRISMA, Preferred Reporting Items for Systematic Reviews and Meta-Analyses.

the healthcare area of the studies included, refined the research stages considered and included more details on the implementation of the evaluated interventions.

### Patients and public involvement
No patients or public were involved in the study.

### RESULTS
The database search yielded 1399 citations after deduplication (see figure 1). Screening of titles and abstracts resulted in a first classification, after which 435 papers were included for full-text review. We also reviewed the full text of 24 additional references found through forward citation searching. Furthermore, a grey literature search yielded seven additional references. Finally, 109 references were included. Some of these interventions appeared in more than one reference and some of the references contained more than one intervention. Ninety of these references

(86 observational and 4 randomised studies) described 11 evaluated interventions and the other 19 (12 research studies, 2 editorials, 2 blogs, 1 commentary, 1 essay and 1 perspective) described 20 non-evaluated interventions. Figure 2 displays these 31 interventions according to their categorisation and the research stage where they can be performed. Moreover, table 1 shows all interventions in a tabular format together with their rationale. All interventions reported in this section were found in the literature and do not necessarily correspond to the personal ideas of the scoping review authors.

Among the 11 evaluated interventions identified, we found a variety of measures used to assess their effect on adherence to RGs, including:
▶ Score for completeness of reporting for each paper, either assigning different or equal weights to RG items, on a 0–10 scale.
▶ Percentage of items reported for each paper.

| TYPE OF INTERVENTION | | EDUCATION | GRANT WRITING | PROTOCOL WRITING | MANUSCRIPT WRITING | MANUSCRIPT SUBMISSION | JOURNAL PEER REVIEW | COPY-EDITING | POST-PUBLICATION |
|---|---|---|---|---|---|---|---|---|---|
| | **TRAINING** on the practical use of RGs | Introduction of RGs & journalology into graduate curricula (18-22); Student's development of research protocols using RGs (21) | Funder's support of author training on RGs (23) | | | | Training for peer reviewers and editors on RGs by journals (22,23) | | |
| | Enhancing **ACCESSIBILITY** and **UNDERSTANDING** | Dissemination of RGs by scientific associations (24); Translation of RGs to further languages (25); Development of expanded databases of examples for each RG (26) | | | | | | | |
| | **ENCOURAGING** adherence | | Author use of RGs as a template for grant applications' proposals (21) | Required checklist for ethics approval application (11) | Author use of the writing aid tool COBWEB (12); Author use of a structured approach for reporting research (47,112); Author markup of the manuscript to indicate where each RG item is addressed (109); Funder's requirement of checklists in author's report (21,108) | Editorial statement endorsing certain RGs (27–46,48–106,113); Recommendation or requirement to follow RGs in the "Instructions to authors" (27–46,48–106,113); Requirement to submit a RG checklist together with the manuscript indicating page numbers corresponding to each item (27–46,48–106,113); Journal development of core versions of RGs containing key items (110); Guidance to authors on manuscript preparation by publication officers (111); Requirement to populate and submit a RG checklist with text from the manuscript (114) | Suggestion for peer reviewers to use RGs (107); Editor's questions to peer reviewers about whether the authors have followed RGs (115) | | |
| | **CHECKING** adherence and providing **FEEDBACK** | | | | | | Completeness of reporting check by editors (117); Peer review against RGs (118); Internal peer review against RGs by a trained editorial assistant (120); Implementation of the automatic tool Statreviewer (121); Email to authors to revise the manuscript according to RGs (13); Implementation of the web tool WebCONSORT (119) | Completeness of reporting check at copy-editing (122) | Post-publication peer review (123) |
| | Involvement of **EXPERTS** | | | Medical writer involvement (108); Statistician involvement (78,128-130) | | | | | |
| | | **EDUCATION** | **GRANT WRITING** | **PROTOCOL WRITING** | **MANUSCRIPT WRITING** | **MANUSCRIPT SUBMISSION** | **JOURNAL PEER REVIEW** | **COPY-EDITING** | **POST-PUBLICATION** |
| | | | **BEFORE STUDY CONDUCT** | | **AFTER STUDY CONDUCT** | | | | |
| | | | | | **RESEARCH STAGE** | | | | |

**Figure 2** Typology of interventions to improve adherence to RGs according to type of intervention and research stage. Evaluated interventions are shown in bold. CONSORT, Consolidated Standards of Reporting Trials; RGs, reporting guidelines.

▶ Percentage of compliance per RG item.
▶ Score for the Manuscript Quality Assessment Instrument[17] for each paper.

Due to the heterogeneity of these measures and for the sake of clarity, we prefer to omit the information on the exact effect sizes in the main body of the manuscript and show it in online supplementary file 2, together with the implementation details of the evaluated interventions. In this way, these effects can be understood in an appropriate context.

Research gaps identified (see figure 3) included the evaluation of interventions (1) on training on the use of RGs and improving understanding of these, (2) at early stages of research (education, grant writing or protocol writing) and (3) after the final acceptance of the manuscript (copyediting or postpublication peer review).

Hereafter, we describe the interventions found for each category (table 1 and online supplementary file 2 summarise these interventions).

### Training on the practical use of RGs

Four non-evaluated interventions related to educating different stakeholders on the practical use of RGs were found.[18–23]

In a first step, health profession schools could incorporate RGs into curricula that address research methodology and publication standards.[18–22] In line with this, students could develop protocols for coursework and research using RGs such as Standard Protocol Items: Recommendations for Interventional Trials (randomised trials) and PRISMA-Protocol (systematic reviews), and educators

may encourage adherence to those guidelines and grade the protocols using them.[21] For their part, funders may consider supporting author training on RGs.[23] Finally, journals or publishers may consider investing resources in training editors and reviewers on the content and use of RGs.[22 23]

### Enhancing accessibility and understanding

We identified three non-evaluated interventions focused on increasing the awareness of the existence of RGs, as well as the authors' understanding of the content of these.[24–26]

First, international scientific associations may play an important role in disseminating and popularising RGs to large audiences.[24] Second, RG developers might consider translating them to new languages that have not been addressed yet.[25] Finally, further databases of examples of good reporting for different RGs that are accessible to authors can be developed, as has been done for CONSORT.[26]

### Encouraging adherence

Fourteen interventions found were associated with different strategies to facilitate compliance with RGs.[11 12 21 27–115] Six of these were evaluated.[12 27–107 113]

Funders might require authors to use RGs as a template for grant application proposals.[21] Later on, research ethics boards may require that protocols submitted for ethical approval clearly state which RGs the study will be using based on the study design, and that RG checklists are part of the application for ethics approval.[11] Funders

**Table 1** Rationale of the interventions identified.

| Group | Intervention | Rationale |
|---|---|---|
| Training on the practical use of RGs | Introduction of RGs and journalology into graduate curricula[18–22] | To introduce good research reporting habits early in young researchers' scientific careers. |
| | Student's development of protocols for coursework and research using RGs[21] | |
| | Funder's support of author training on RGs[23] | Authors, editors and peer reviewers have insufficient training in issues related to reporting. |
| | Training for peer reviewers and editors on RGs by journals[22 23] | |
| Enhancing accessibility and understanding | Dissemination of RGs by scientific associations[24] | A large number of researchers are not aware of the existence of RGs. |
| | Translation of RGs to further languages[25] | Language barriers may affect the proper use of RGs. |
| | Development of expanded database of examples for each RG[26] | Authors need more examples of good reporting to properly understand certain items. |
| Encouraging adherence | Author use of RGs as a template for grant application proposals[21] | Using RGs in early stages may facilitate completeness of reporting of published research. |
| | Required checklist for ethics approval application[11] | |
| | Funder's requirement of checklists in author's report[21 108] | |
| | Author use of the writing aid tool COBWEB[12] | (A) Authors need help to successfully adhere to RGs at the writing stage and (B) Dividing RG items into bullet points and providing examples might help. |
| | Author use of a structured approach for reporting research[47 112] | (A) To help authors avoid omissions, (B) to aid reviewers and editors in appraising articles and (C) to allow more efficient data extraction during the systematic review process. |
| | Author mark-up of the manuscript to indicate where each RG item is addressed[109] | |
| | Editorial statement endorsing certain RGs[27–46 48–106 113] | Authors read editorial statements and follow 'Instructions to authors'. |
| | Recommendation or requirement to follow RGs in the 'Instructions to authors'[27–46 48–106 113] | |
| | Requirement to submit an RG checklist together with the manuscript indicating page numbers corresponding to each item[27–46 48–106 113] | Authors may not consider editorial statements or recommendations in 'Instructions to authors' to be important. Compulsory submission of checklists or text mark-up may encourage authors to be more compliant with RGs. |
| | Requirement to populate and submit an RG checklist with text from the manuscript[114] | |
| | Journal development of core versions of RGs containing key items[110] | Focusing on the most important items could be more effective than considering the whole checklist. |
| | Guidance to authors on manuscript preparation by publication officers[111] | Trained journal officers may enhance authors' compliance with RGs during manuscript preparation. |
| | Suggestion for peer reviewers to use RGs[107] | Peer reviewers often do not detect reporting flaws. Therefore, they may need to follow a more systematic approach and use RGs. |
| | Editor's questions to peer reviewers about whether the authors have followed RGs[115] | |
| Checking adherence and providing feedback | Completeness of reporting check by editors[117] | Requiring checklists at submission does not guarantee adherence. Editors and peer reviewers have to check whether submitted papers are compliant with RGs. |
| | Peer review against RGs[118] | |
| | Internal peer review against RGs by a trained editorial assistant[120] | It is extremely unlikely that the average clinical peer reviewer has the methodological expertise to check a paper against RGs. |
| | Implementation of the automatic tool Statreviewer[121] | |
| | Email to authors to revise the manuscript according to RGs[13] | It might be more effective to ask authors for adherence to RGs during the revision process because they will do anything to get their paper published. |
| | Implementation of the tool WebCONSORT[119] | |
| | Completeness of reporting check at copyediting[122] | Copyediting and postpublication offer alternate time points to improve adherence to RGs. |
| | Postpublication peer review[123] | |
| Involvement of experts | Statistician involvement (78 128–130) | Professionals with specific knowledge of RGs might help authors when designing, conducting or reporting their research. |

Medical writer involvement.[108]
COBWEB, CONSORT-based web tool; CONSORT, Consolidated Standards of Reporting Trials.
RGs, reporting guidelines.

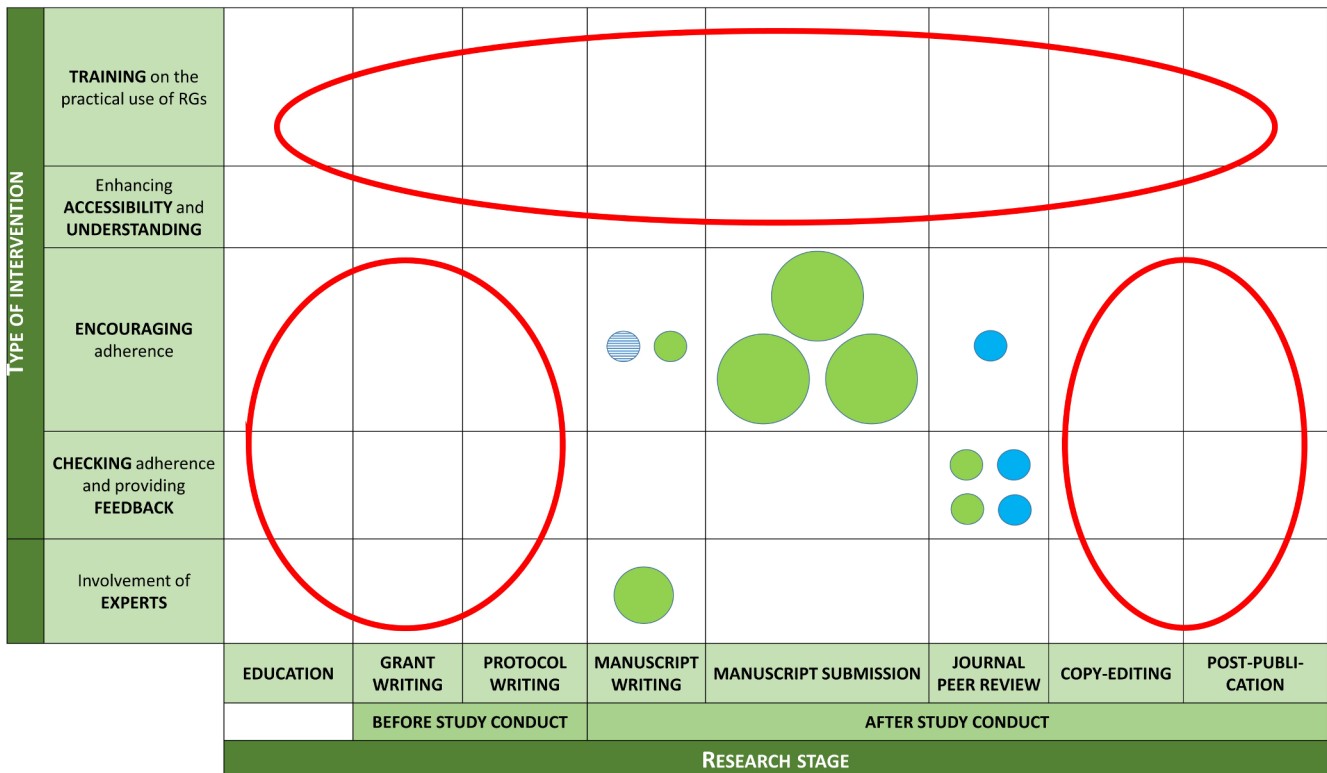

**Figure 3** Gaps in research on the evaluation of interventions to improve adherence to RGs. Each circle represents one intervention. Variables displayed: (1) Circle size: number of studies evaluating each intervention (bigger=more studies); (2) Circle colour: study design of those studies (blue for RCTs and green for observational studies) and (3) Circle fill: kind of RG implementation (plain for checklist and stripes for bullet points and examples). Research gaps are highlighted in red. RCTs, randomised controlled trials; RGs, reporting guidelines.

could also encourage adherence to RGs by asking for RG checklists as part of the authors' report.[21 108]

One initiative to support authors adhering to RGs at the writing stage of the manuscript has been COBWEB, a writing aid tool that aims to help authors adequately combine the different extensions of the CONSORT statement.[12] This tool divided the CONSORT items into bullet points showing the key elements that need to be reported together with examples of adequate reporting. The impact of COBWEB was evaluated in a randomised trial that showed a large effect of this intervention[12] (see online supplementary file 2 for more details about this and other evaluated interventions). A second option to support authors at manuscript writing is that they follow a more structured approach. For example, ClinicalTrials.gov requires authors to report key information in a tabular format when registering a study or making available its results.[116] This has been shown to be effective: some results posted on this platform, especially harms, are more complete than those in corresponding journal articles reporting the same trials.[47] Another possibility to improve the structure of manuscripts is to include new subheadings corresponding to different RG items within the traditional format introduction, methods, results and discussion, as the American Journal of Orthodontics and Dentofacial Orthopedics (AJO-DO) proposed.[112] Finally, authors may also avoid omissions when writing the

manuscript if mark-up the text and highlight where each item of the relevant checklist is addressed.[109]

At the manuscript submission stage, different editorial actions have been taken to improve adherence to RGs. The most popular is what has traditionally been defined as journal endorsement of RGs, which is usually defined as one or more of the three following interventions: (1) journal editorial statement endorsing certain RGs; (2) requirement or recommendation in journal's 'Instructions to Authors' to follow certain RGs when preparing their manuscript or (3) requirement for authors to submit the appropriate RG checklist together with their manuscript indicating page numbers corresponding to each item.[6] Dozens of observational studies have explored the possible effect of journal endorsement of different RGs in different clinical areas.[27–46 48–106 113] A recent systematic review focused on CONSORT evaluations showed relative but suboptimal improvements in the completeness of reporting in journals by following the aforementioned policies,[6] while another systematic review considering nine other guidelines showed no improvements.[3]

Journals might also consider other strategies to enhance adherence to RGs at submission. A first option could be to develop shorter, core versions of RGs containing key items, which could be provided to authors as part of the submission process.[110] Second, they might introduce publication officers in order to provide guidance

to authors on preparing manuscripts for submission.[111] Third, editors may ask authors to populate the relevant checklist with text from their manuscript and not accept a submission unless this is provided.[114]

Finally, editors may suggest that peer reviewers use RGs.[107] In addition, by asking peer reviewers questions about whether the author has followed RGs, this might be an indirect way to encourage them.[115]

### Checking adherence and providing feedback

Eight interventions were related to monitoring level of compliance with RGs of the manuscripts and providing instructions to authors on how to improve the reporting of missing or incorrect items.[13 117–123] Four of them were evaluated.[13 117–119]

Some journals have opted for implementing RGs at peer review. First, an associate editor may assess manuscripts for adherence to the relevant RG and ask authors to make changes accordingly.[117] This process may be repeated until the associate editor thinks that the manuscript can move to the next step of the review process, leading to an editorial decision. This intervention was evaluated at the AJO-DO and showed satisfactory results: 33 of 37 items reached perfect compliance.[117] Second, peer reviewers could also assess the manuscripts against the appropriate checklist.[118] While the observed effect of this intervention was slightly positive, it was smaller than hypothesised. In fact, investigators pointed out that authors tended to comply better with suggestions coming from standard reviews rather than from reviews against RGs, implying that it might be difficult to adhere to high methodological standards at late stages of research if these standards are not considered earlier in the research process. Third, journals could also ask trained editorial assistants to check manuscripts against RGs[120] or to implement automatic peer-review tools such as StatReviewer,[124] software that automatically checks adherence to RGs and evaluates the appropriate use and reporting of statistical tests.[121] Currently, its performance is being assessed through a pilot trial in collaboration with four *BioMed Central Journals*.[121] In any of those cases, emails could be sent to authors asking them to revise the manuscript according to guidelines.[13] To do this, the EQUATOR network has provided standard letters that can be used (1) after checks by an editor or a single peer reviewer, (2) after full peer review or (3) alongside acceptance.[125] Furthermore, at the time of author revision of the manuscript, Hopewell *et al* found no significant effect when incorporating WebCONSORT, a web-based tool that generates a unique list of items customised to the trial design, to the revision process of journals that endorsed CONSORT but had no active policy for implementing it.[119] Finally, in a late stage of the publication process, copyediting of the manuscript could also ensure that all items are covered.[122]

Once the paper is published, the scientific community could use online platforms of postpublication peer review such as PubPeer[126] or ScienceOpen[127] to evaluate the adherence to RGs of published articles and to provide feedback to authors.[123]

### Involvement of experts

Two interventions identified implied interaction and cooperation between authors and experts on methodology and reporting at different stages of research.[78 108 128–130] One of them was evaluated.[78 128–130]

On the one hand, statisticians (or epidemiologists or other quantitative methodologists) may get involved in the design, conduct or reporting of the study might contribute to properly reporting key areas such as sample size calculation, randomisation, blinding and appropriate statistical analysis.[129] While three studies found a statistically significant positive relationship between CONSORT scores and statistician involvement,[78 129 130] another one did not.[128] On the other hand, it has been hypothesised that the involvement of medical writers during the manuscript writing stage of research could improve the completeness of reporting.[108]

### Interventions described in papers coauthored by authors of this scoping review

Twenty-five (of 109) included references describing 21 (of 31) included interventions were coauthored by at least one of the authors of this scoping review.[12 13 20–23 26 47 54 55 63 67 74 76 80 104 107 111 114 115 117–120 123]

### DISCUSSION

In this scoping review, we identified 31 interventions to improve adherence to RGs. We have also determined the gaps in research on the evaluation of this type of interventions. By considering a wide range of RGs as well as their extensions and merging the evidence found in the published and grey literature, this review provides a broad picture of how the problem of enhancing adherence to RGs has been tackled so far and could be faced in the future.

This study reveals that most published research aimed at improving adherence to RGs has been conducted in journals. Typically, journal strategies range from making available editorial statements that endorse certain RGs, recommending or requiring authors to follow RGs in the 'Instructions to authors', and requiring authors to submit an RG checklist together with the manuscript, with page numbers indicated for each item. However, these strategies have been shown not to have the desired effect.[3 6 131] Recent research has called for more active and enforced journal policies throughout the editorial process, such as requiring the use of structured approaches with new subheadings adapted to different kinds of study designs,[112] which was also found to be beneficial in a new study outside of our search period[132]; providing guidance on manuscript preparation[111]; making sure the peer-review process involves editorial assistants who have specific training on reporting issues[120] and implementing automatic peer-review tools.[121] Journals will vary in their ability

to make some of these strategies effective, depending on factors such as their resources, their guidelines to peer reviewers and the dedication of their editors—many editors and editorial staff work part time and have a limited amount of time.

Moreover, editors' education and performance should be improved. A recent study pointed out that more than one-third (39%) of the manuscripts classified as randomised trials by the editorial staff were not actually randomised trials.[119 133] Consequently, it seems difficult to improve author and peer reviewer adherence to RGs if journal gatekeepers are not properly trained in methodological and reporting issues.

Apart from journals, editors and peer reviewers, other key stakeholders such as medical schools, research funders, universities and other research institutions should also take responsibility regarding this issue. This scoping review provides some strategies to follow. However, as the problem is complex and the possible interventions are varied, enhancing the completeness of reporting most likely depends not so much on any isolated action but on a set of strategies by several different stakeholders. These could be enacted at different stages of research, from education to article postpublication.

For interventions aiming to improve adherence to RGs, we should require the same level of evidence that we require for interventions to improve health. For this reason, it is striking that we found only four published randomised trials that evaluated interventions to improve adherence to RGs.[12 107 118 119] Among these trials, statistically significant effect of the intervention was only observed for the use of the writing aid tool for authors COBWEB.[12] While performing an additional review against RGs showed slightly positive but not significant effect,[118] suggesting the use of RGs to peer reviewers[107] or implementing at the process of author revision of the manuscript the web-based tool WebCONSORT showed no benefit.[119] The rest of the evaluations of interventions found (86 of 90) were observational studies, whose results are subject to the influence of confounding factors. As already mentioned, the impact of journal endorsement on completeness of reporting was suboptimal.[3 6] However, completeness of reporting improved remarkably when RGs were actively implemented by editors (eg, if editors perform a completeness of reporting check of the manuscript[117]) and when research results were posted in a tabular format without discussion or conclusions.[47] Future randomised trials should consider evaluating these interventions or addressing some of the research gaps identified in this review, such as improving adherence to RGs at the grant application or protocol writing stages.

A few of the interventions found in this review were shown to enhance adherence to RGs. However, it is noteworthy there is no evidence that some successful interventions[12 117] have been implemented more widely later. For this reason, more resources and efforts are needed to further implement these interventions in other settings, evaluate the effect, and share the results with the scientific community. In any case, it is important to keep in mind that contemporary publication culture may harm the potential improvements in reporting quality. This could result from the fact that most scientists feel that the primary evaluation tool of their research is the quantity of their scientific output rather than its quality[134]; and such attitudes may undermine the potential effect of any intervention to improve adherence to RGs.

Our scoping review has some limitations. First, we did not formally assess the methodological quality of the studies that evaluated interventions. Second, restricting to certain databases or not having standard search terms for the databases searched may have excluded relevant publications. Third, it is possible that we could have missed evidence of possible interventions that may have never been reflected in the published or grey literature but are instead used in practice and continue to be used. For example, journals might be applying specific editorial strategies that are not publicly available on their websites or in the published literature.

## CONCLUSION

Improving adherence to RGs is one of the key issues in order to enhance complete and accurate reporting and therefore reduce waste in research.

Different stakeholders—such as research funders, ethics boards and journals—should consider implementing and evaluating some of the interventions identified in this study.

**Acknowledgements** The authors thank the MiRoR Project (http://miror-ejd.eu/) and Marie Sklodowska-Curie Actions for their support. The authors also thank Matt Elmore for editorial help. This review is part of a larger project whose next goals are (1) to capture editors' perceptions on the barriers and facilitators of some promising interventions identified in this review, (2) to explore new possible interventions and (3) to evaluate one of these interventions in collaboration with *BMJ Open*.

**Contributors** All authors contributed to conceptualising and designing the study. DB, EC and JJK independently performed screening. DB and JJK independently performed data extraction. DB performed initial data synthesis and EC, IB, DM, DA and JJK refined it. DB drafted the manuscript. EC, IB, DM, DA and JJK made major revisions. Due to the strong involvement of JJK and EC at several different stages of the study, all authors agreed to consider them joint senior authors of the scoping review, although EC was the only senior author of the protocol. All authors read and approved the final manuscript, which was completed in April 2018. DA passed away in June 2018 and therefore could not approve the revised manuscript (November 2018).

**Funding** This scoping review belongs to the ESR 14 research project from the Methods in Research on Research (MiRoR) project (http://miror-ejd.eu/), which has received funding from the European Union's Horizon 2020 research and innovation programme under the Marie Sklodowska-Curie grant agreement No 676207. DM is supported through a University Research Chair (University of Ottawa).

**Competing interests** DA and DM are directors of the UK and Canadian EQUATOR Centres, respectively. IB is deputy director of French EQUATOR Centre.

**Patient consent for publication** Not required.

**Provenance and peer review** Not commissioned; externally peer reviewed.

**Data sharing statement** The datasets used and/or analysed during the current study are available from the corresponding author on reasonable request.

and indication of whether changes were made. See: https://creativecommons.org/licenses/by/4.0/.

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
