## [Reviewer comments · BMJ Open]

ARTICLE DETAILS

TITLE (PROVISIONAL)	A scoping review on interventions to improve adherence to reporting guidelines in health research
AUTHORS	Blanco, David; Altman, Doug; Moher, David; Boutron, Isabelle; Kirkham, Jamie; Cobo, Erik

VERSION 1 - REVIEW

REVIEWER	Mario Malicki Faculty of Health, Hogeschool van Amsterdam, Amsterdam, Netherlands
REVIEW RETURNED	04-Oct-2018

GENERAL COMMENTS	Dear Authors, Thank you for the opportunity to review this manuscript. I enjoyed reading it, and find RG dear to my heart, but I feel your paper needs improving: 1. With sadness of Doug passing, please indicate in acknowledgments or where the editor prefers about his passing, and did he indeed approve the final version, or any changes that occurred in the last few months of the paper in light of your submission date. Please also indicate why there has been a change from the order of the authors in the protocol and current publications, and was this order approved by Doug or not.2. Please rename or clarify categories of interventions in the abstract: collaboration among authors and experts – on what? Improving adherence or? Training – on? Subsequently I advise completely renaming the categories in your review: 1) collaboration to Involvement of experts, 2) Training to Education on RG 3)improving understanding – implies some before-after knowledge testing and improving of knowledge of users, but the interventions described are rather improving the materials surrounding RG – and that way enhancing accessibility and self-learning or self-help of users. – so perhaps Developing training materials or something of the sort is a better term. 4) Monitoring should be renamed to Checking adherence, as including providing feedback only in this one – indirectly implies feedback is not provided anywhere else.3. If you mention in the abstract that the effect of interventions was analyzed, do mention the found effect of this in the results4. Only 4 of the interventions found had been evaluated by randomized trials – this does not seem important for the overall picture – if you want to imply that the quality of testing
---

interventions was bad, please specify so. But as you state you did not analyse the quality of the studies or the interventions – best refrain from stating so.

5. Please leave out the future projects' goals from the abstract's conclusion

6. allows us to provide a broad picture – please use past tense – the work is finished

7. The paper that you cite for ~85% of research waste - clearly states “A precise quantification of the amount of waste in the reporting of research is not possible.” And yet u use this review paper to say this – please explain or provide other data that mentions this approximate number.

8. We aim to analyse – aimed

9. Please provide detailed search results per each database. Please also specify how were the duplicates removed – automatically by Mandalay or reference comparison by the authors. Also indicate the number of duplicates. Additionally, based on your search that seems to have been adopted to other databases this seems to have been more of a systematic instead of a scoping review. Lastly, as you had the search strategies in your protocol, there is no need to report them here unless there were changes, so please refer to them in the protocol.

10. Please consider adding QUOROM description in Supp materials 1

11. PRISMA Extension for Scoping Reviews has been published – and you mention in you protocol – perhaps you should use it.

12. Please explain why Email to authors to revise the manuscript and webs consort tool does not belong under journal peer review process – having an internal peer review process that then finds non-adherence, will also require the authors to revise the paper – the way you posed this seem to imply that journals should focus on complete reporting only after acceptance of the paper – and not requiring adherence in submission phase. If so , I still would merge the two categories and make a journal stage category – as internal check can be pre-peer review, parallel to it, or only after it has been decided it would be accepted.

13. You included in the interventions – post publication peer review – it would be good to mention in the paper regarding your definitions – does adherence then include corrections made following the publication of the paper – or are the implementation and complete reporting meant only with the publication of the first version of the paper, and a comment on the stage for preprints in light of that is also welcome.

14. Please remove the mention of the goal of the overall project from the conclusion – as that is not the conclusion of your review – additionally – you state: “For example, a decrease in waste of research from 85% to 70% would double the output of valuable research” – do you have proof about this, If not I suggest you tone it down greatly.

15. Importantly, you state “we present a brief description of the interventions found for each category”. However, the subtitles below read more like an opinion paper than a review – Please do not state what could be done, do so in the conclusions or as the last paragraph of these subsections, rather state what your review has found – what and how many interventions were suggested which attempted what, and which ones were not suggested but you feel could also be done within those respective categories. Descriptive stat data on the effect of their interventions is also welcome, or at least expand the supplementary table 3 with measurements, not only mentioning significant difference found or

	not. A comment on the measures and variables used to assess adherence would also be highly appreciated. 16. Please have a subsection on any deviations from your protocol (if there were or state there were not) – there you stated you would quantify the effect of the evaluated studies and list in which healthcare area they were evaluated or suggested? – I feel this has not been truly addressed. 17. Finally, as some of the included interventions or suggestions were from the authors of this review – please indicate were there differences in how those studies/abstracts were screened or assessed, and indicate the number and characteristics of those studies appropriately in the text and tables. I hope my comments can help you improve your manuscript, Kind regards, Mario Malicki
--	---

REVIEWER	Vedran Katavic University of Zagreb School of Medicine, Croatia
REVIEW RETURNED	05-Oct-2018

GENERAL COMMENTS	A thoughtful approach to an interesting and important topic. Executed really well!
---

REVIEWER	Howraman Meteran Respiratory Research Unit, Bispebjerg University Hospital, Copenhagen, Denmark
REVIEW RETURNED	14-Oct-2018

GENERAL COMMENTS	Thanks for your submission. The manuscript (review) is a part of a larger project. The overall goal of the larger project is highly relevant and the need of such work is warranted. In my opinion, the authors are doing right in separating the different parts of the larger project. The aims of the study are clearly described, accordingly searched for and scrutinised in an understandable manner and the review can be accepted for publication.
---

VERSION 1 – AUTHOR RESPONSE

Reviewers' Comments to Author:

Reviewer: 1

Reviewer Name: Mario Malicki

1. With sadness of Doug passing, please indicate in acknowledgments or where the editor prefers about his passing, and did he indeed approve the final version, or any changes that occurred in the last few months of the paper in light of your submission date. Please also indicate why there has been

a change from the order of the authors in the protocol and current publications, and was this order approved by Doug or not.

All these points have been addressed in the Acknowledgements section. The final version of the manuscript was approved by all authors in April 2018 and submitted to another journal which could not find suitable reviewers for it in the next few months. After that, we submitted it to BMJ Open without making changes.

2. Please rename or clarify categories of interventions in the abstract: collaboration among authors and experts – on what? Improving adherence or? Training – on?

Subsequently I advise completely renaming the categories in your review: 1) collaboration to Involvement of experts, 2) Training to Education on RG 3)improving understanding – implies some before-after knowledge testing and improving of knowledge of users, but the interventions described are rather improving the materials surrounding RG – and that way enhancing accessibility and self-learning or self-help of users. – so perhaps Developing training materials or something of the sort is a better term. 4) Monitoring should be renamed to Checking adherence, as including providing feedback only in this one – indirectly implies feedback is not provided anywhere else.

The categories of interventions have been renamed to (1) Training on the use of reporting guidelines, (2) Enhancing accessibility and understanding, (3) Encouraging adherence, (4) Checking adherence and providing feedback, and (5) Involvement of experts. Further details on what kind of interventions each category comprises can be found in Lines 203-211. Intervention “Dissemination of RGs by scientific associations” was subsequently moved from category (3) to (2).

3. If you mention in the abstract that the effect of interventions was analyzed, do mention the found effect of this in the results

Please see our response to your comment number 15.

4. Only 4 of the interventions found had been evaluated by randomized trials – this does not seem important for the overall picture – if you want to imply that the quality of testing interventions was bad, please specify so. But as you state you did not analyse the quality of the studies or the interventions – best refrain from stating so.

We have removed that sentence from the abstract.

5. Please leave out the future projects’ goals from the abstract’s conclusion

We have removed that piece of information from the abstract.

6. allows us to provide a broad picture – please use past tense – the work is finished

That verb has been changed to past tense (line 47).

7. The paper that you cite for ~85% of research waste - clearly states “A precise quantification of the amount of waste in the reporting of research is not possible.” And yet u use this review paper to say this – please explain or provide other data that mentions this approximate number.

The reference we used was wrong. We have included the right reference where that number comes from (line 56).

8. We aim to analyse – aimed

That verb has been changed to past tense (line 108).

9. Please provide detailed search results per each database. Please also specify how were the duplicates removed – automatically by Mandalay or reference comparison by the authors. Also indicate the number of duplicates. Additionally, based on your search that seems to have been adopted to other databases this seems to have been more of a systematic instead of a scoping review. Lastly, as you had the search strategies in your protocol, there is no need to report them here unless there were changes, so please refer to them in the protocol.

We added information on how the duplicates were removed (lines 135-136), and we included the number of duplicates in the flow diagram (Figure 1). Unfortunately, we did not record the exact number of references for each database - the Joanna scoping review manual we were using did not specify to do so - and our subscription to EMBASE has already expired.

On the other hand, as there were not changes in the search strategies, we deleted the supplementary file where we reported one of them and just referred to the protocol as you suggested (lines 133-134). In our opinion, scoping review is the right term for this type of study, since we focused on a broad question, considered a very broad range of study types (including commentaries, letters...), performed a grey literature search, identified research gaps, and carried out other typical actions for scoping reviews.

10. Please consider adding QUOROM description in Supp materials 1

QUOROM has been added to Supplementary file 1 and this file 1 has been renamed to “Description of the acronyms and full names of all reporting guidelines considered” so that it could contain QUOROM and not only the RGs shown on the EQUATOR website as “Reporting Guidelines for main study types”.

11. PRISMA Extension for Scoping Reviews has been published – and you mention in you protocol – perhaps you should use it.

This was also suggested by the handling editor. As the Word version of the checklist is still not available, we have asked the handling editor on how to proceed. Following their suggestion, we have created our own checklist in Word format (based on the E&E Document), completed it and uploaded it together with the manuscript.

12. Please explain why Email to authors to revise the manuscript and webs consort tool does not belong under journal peer review process – having an internal peer review process that then finds non-adherence, will also require the authors to revise the paper – the way you posed this seem to imply that journals should focus on complete reporting only after acceptance of the paper – and not requiring adherence in submission phase. If so , I still would merge the two categories and make a journal stage category – as internal check can be pre-peer review, parallel to it, or only after it has been decided it would be accepted.

The two categories have been merged under “Journal peer review”.

13. You included in the interventions – post publication peer review – it would be good to mention in the paper regarding your definitions – does adherence then include corrections made following the publication of the paper – or are the implementation and complete reporting meant only with the publication of the first version of the paper, and a comment on the stage for preprints in light of that is also welcome.

We have modified our definition adherence accordingly: “Action(s) taken by authors to ensure that a research report is compliant with the items recommended by the appropriate/relevant reporting guideline. These can take place before or after the first version of the manuscript is published” (Box 1).

14. Please remove the mention of the goal of the overall project from the conclusion – as that is not the conclusion of your review – additionally – you state: “For example, a decrease in waste of research from 85% to 70% would double the output of valuable research” – do you have proof about this, If not I suggest you tone it down greatly.

The description of the follow-up projects has been moved to the final paragraph of the Discussion section. Additionally, the sentence you mention has been removed and the last paragraph of the Conclusion section has been modified.

15. Importantly, you state “we present a brief description of the interventions found for each category”. However, the subtitles below read more like an opinion paper than a review – Please do not state what could be done, do so in the conclusions or as the last paragraph of these subsections, rather state what your review has found – what and how many interventions were suggested which attempted what, and which ones were not suggested but you feel could also be done within those respective categories. Descriptive stat data on the effect of their interventions is also welcome, or at least expand the supplementary table 3 with measurements, not only mentioning significant difference found or not. A comment on the measures and variables used to assess adherence would also be highly appreciated.

The first part of the results section has been rewritten, including a comment on the measures used to assess adherence as well as a justification of why we believe that it is more appropriate to include the exact effect sizes in Supplementary file 2 rather than in the main body of the manuscript (lines 236-256). Furthermore, we rewrote the last column of Supplementary file 2 and, instead of mentioning significant/Not significant, we specified the precise effect size of each intervention on completeness of reporting.

Furthermore, for each category of interventions, we included in the beginning of each section a paragraph with the exact number of evaluated and non-evaluated interventions found, as well as their references, in order to make it clearer to the readers. Regarding the rest of these sections, we would like to keep using the modal verbs we used (“could”, “might”) for the suggested interventions because they were usually presented using that suggestive format in the original papers. For the interventions that were implemented and evaluated, we tried to show in the text that they are not merely suggestions but actions that were actually taken and assessed. We believe that, if readers want to access a concise summary of the interventions found, they can have it in Table 1 or Figure 2 – this has been emphasised in Lines 236-238, 255-256. However, some parts of these sections were slightly modified to make the text clearer.

16. Please have a subsection on any deviations from your protocol (if there were or state there were not) – there you stated you would quantify the effect of the evaluated studies and list in which healthcare area they were evaluated or suggested? – I feel this has not been truly addressed.

We have included a subsection on deviations from the protocol at the end of the Methods section. Regarding the quantification of the effects, please see our response to comment number 15.

17. Finally, as some of the included interventions or suggestions were from the authors of this review – please indicate were there differences in how those studies/abstracts were screened or assessed, and indicate the number and characteristics of those studies appropriately in the text and tables.

A paragraph on how these studies were screened and assessed was added at the end of Data Extraction section. Moreover, a subsection indicating some details of these studies, as well as their references, was added at the end of Results section.

VERSION 2 – REVIEW

REVIEWER	Mario Malicki Faculty of Health Amsterdam University of Applied Sciences Amsterdam, the Netherlands
REVIEW RETURNED	05-Dec-2018

GENERAL COMMENTS	Dear Authors, First, I would like to say thank you for making all of the extensive changes I previously asked for. I find the current version showcases your research in a much nice way then the previous one, and would like to suggest only a few more minor amends: 1) Abstract – Research gaps identified included the evaluation of interventions – Additionally, we identified lack of evaluated interventions on2) Abstract - and improving understanding of these – improving their understanding3) Abstract – This scoping review identifies – identified4) Abstract - Future randomised trials should consider evaluating some of the interventions that have not been assessed yet, therefore addressing the research gaps identified. – Additional research is needed to assess effectiveness of many of these interventions.5) Supplementary table 2 – U state: **As the 80 individual studies that belong to this category used different measures of adherence to reporting guidelines, we report here the measures used in the two systematic reviews that summarized the pooled results of most of these studies (3,6). However it is not clear which measures belong to the studies 3, and 6 as you only reference study 6 in the table – please indicate with ** - which of the measurements are those you took from the SR, and which are the ones you compiled. Furthermore, please explain in methods section how you meta-analysed or summarized results from observational studies that were not included in the SRs. As 86 observational studies were conducted, of which as you say 9+14 were before and after, a more detailed summary of these results would be appreciated in the discussion section, and not just a reliance on RCTs.6) Discussion - This study reveals that it is primarily journals that have made most of the efforts to improve adherence to reporting guidelines in health research – although they can certainly do more. – I would suggest rephrasing this – What could be claimed is that most research published for improving adherence was conducted in journals – not that educational interventions have done less.7) Discussion - The rest of the evaluations of interventions found (86 of 90) were observational studies, whose results are subject to the influence of confounding factors (6). – While confounding factors do play a role, as you did have 86 studies – please do mention a summary results of those in the discussion – do they indicate journals actions can lead to improvements of reporting – albeit small ones, or not? Finally, it seems odd that you only cite ref 6 as a paper for discussion on the value of observational studies in medicine – either remove the reference or add additional studies on that extensive topic, especially in light of increasing use of obs studies in meta-analyses.8) Again, I would ask you remove : This review is part of a larger project whose next goals are (i) to capture editors' perceptions on
---

	the barriers and facilitators of some promising interventions identified in this review, (ii) to explore new possible interventions, and (iii) to evaluate one of these interventions in collaboration with BMJ Open. From discussion, and include this information in acknowledgments. 9) Finally, while you have truly addressed all of the other issues, and stated: we would like to keep using the modal verbs we used (“could”, “might”) for the suggested interventions because they were usually presented using that suggestive format in the original papers.- the reason I have asked you originally to rephrase this is, it is not clear when one reads these sections are these your suggestions – or have they been proposed by others – and I think this distinction should be made very clear. While many of the paragraphs end with references and possibly indicate previous suggestions, an emphasis on when an idea is your personal observation/idea is welcome. Or in cases there are always previous suggestions, then perhaps mention in the opening paragraph of results that all suggestions reported are those already published. In hopes may comments can further improve your manuscript, Kind regards, Mario Malicki
--	---

VERSION 2 – AUTHOR RESPONSE

1) Abstract – Research gaps identified included the evaluation of interventions – Additionally, we identified lack of evaluated interventions on

Done

2) Abstract - and improving understanding of these – improving their understanding

Done

3) Abstract – This scoping review identifies – identified

Done

4) Abstract - Future randomised trials should consider evaluating some of the interventions that have not been assessed yet, therefore addressing the research gaps identified. – Additional research is needed to assess effectiveness of many of these interventions.

Done

5) Supplementary table 2 – U state: **As the 80 individual studies that belong to this category used different measures of adherence to reporting guidelines, we report here the measures used in the two systematic reviews that summarized the pooled results of most of these studies (3,6). However it is not clear which measures belong to the studies 3, and 6 as you only reference study 6 in the table – please indicate with ** - which of the measurements are those you took from the SR, and which are the ones you compiled. Furthermore, please explain in methods section how you meta-analysed or summarized results from observational studies that were not included in the SRs. As 86 observational studies were conducted, of which as you say 9+14 were before and after, a more

detailed summary of these results would be appreciated in the discussion section, and not just a reliance on RCTs.

- Study 3 reference was missing in the last column – we have referenced it accordingly.
- We have indicated with ** in the second last column that the two measures we report here are the ones used in the two systematic reviews.
- We have mentioned in the end of the Data Synthesis section that no meta-analysis was performed with the observational studies that were not included in the SRs: “We did not perform a meta-analysis of the observational studies assessing journal endorsement of reporting guidelines that were not included in the two systematic reviews previously mentioned (3,6). We considered that, for the purpose of this scoping review, these systematic reviews provided a reliable picture of the impact of this editorial intervention.”
- Regarding the last point of your comment, please see the response to Comment 7).

6) Discussion - This study reveals that it is primarily journals that have made most of the efforts to improve adherence to reporting guidelines in health research – although they can certainly do more. – I would suggest rephrasing this – What could be claimed is that most research published for improving adherence was conducted in journals – not that educational interventions have done less.

Done – sentence changed to “This study reveals that most published research aimed at improving adherence to reporting guidelines has been conducted in journals”.

7) Discussion - The rest of the evaluations of interventions found (86 of 90) were observational studies, whose results are subject to the influence of confounding factors (6). – While confounding factors do play a role, as you did have 86 studies – please do mention a summary results of those in the discussion – do they indicate journals actions can lead to improvements of reporting – albeit small ones, or not? Finally, it seems odd that you only cite ref 6 as a paper for discussion on the value of observational studies in medicine – either remove the reference or add additional studies on that extensive topic, especially in light of increasing use of obs studies in meta-analyses.

- We have made a reference to the results of the observational studies in lines 437-443.
- The reference has been removed.

8) Again, I would ask you remove : This review is part of a larger project whose next goals are (i) to capture editors’ perceptions on the barriers and facilitators of some promising interventions identified in this review, (ii) to explore new possible interventions, and (iii) to evaluate one of these interventions in collaboration with BMJ Open. From discussion, and include this information in acknowledgments.

Done – This sentence has been moved to Acknowledgements.

9) Finally, while you have truly addressed all of the other issues, and stated: we would like to keep using the modal verbs we used (“could”, “might”) for the suggested interventions because they were usually presented using that suggestive format in the original papers.- the reason I have asked you originally to rephrase this is, it is not clear when one reads these sections are these your suggestions – or have they been proposed by others – and I think this distinction should be made very clear. While many of the paragraphs end with references and possibly indicate previous suggestions, an emphasis on when an idea is your personal observation/idea is welcome. Or in cases there are always previous suggestions, then perhaps mention in the opening paragraph of results that all suggestions reported are those already published.

Done – A sentence was added in the final part of the opening paragraph of the results section: “All interventions reported in this section were found in the literature and do not necessarily correspond to the personal ideas of the scoping review authors.”

VERSION 3 - REVIEW

REVIEWER	Mario Malicki Faculty of Health Amsterdam University of Applied Sciences Amsterdam, the Netherlands
REVIEW RETURNED	31-Jan-2019

GENERAL COMMENTS	Dear Authors, thank you for making all of the asked changes. I find the paper now acceptable for publication. Best, Mario
--